# FISHNETS: INFORMATION-OPTIMAL, SCALABLE AGGREGATION FOR SETS AND GRAPHS

## ABSTRACT

Set-based learning is an essential component of modern deep learning and network science. Graph Neural Networks (GNNs) and their edge-free counterparts DeepSets (DS) have proven remarkably useful on ragged and topologically challenging datasets. The key to learning informative embeddings for set members is a specified aggregation function, usually a sum, max, or mean. We propose Fishnets, an aggregation strategy for learning information-optimal embeddings for sets of data for both Bayesian inference and graph aggregation. We demonstrate that i) Fishnets neural summaries can be scaled optimally to an arbitrary number of data objects, ii) Fishnets aggregations are robust to changes in data distribution, unlike standard deepsets, iii) Fishnets saturate Bayesian information content and extend to regimes where MCMC techniques fail and iv) Fishnets can be used as a drop-in aggregation scheme within GNNs. We show that by adopting a Fishnets aggregation scheme for message passing, GNNs can achieve state-of-the-art performance versus architecture size on benchmark datasets over existing architectures with a fraction of learnable parameters and faster training time.

## 1 INTRODUCTION

Aggregating information from independent data in an optimal way is a fundamental problem in statistics and machine learning. On one hand, frequentist analyses need optimal estimators for data compression, while on the other Bayesian analyses need small informative summaries for simulation-based inference (SBI) schemes (Cranmer et al., 2020). In a deep learning context graph neural networks (GNNs) rely on aggregation schemes to pool information over large data structures, where each feature might be weakly informative, but at a graph level might contribute a lot of information for predictive or regression tasks (Zhou et al., 2020).

Up until now, graph aggregation schemes have relied on simple, fixed operations such as mean, max, sum, (Kipf & Welling, 2017; Hamilton et al., 2017; Xu et al., 2019), variance, or trainable variants of these aggregators (Battaglia et al., 2018; Li et al., 2020), which are susceptible to generalisation issues in heterogeneous data aggregation. We introduce a new optimal aggregation scheme grounded in information-theoretic principles. By leveraging the additive structure of the log-likelihood for independent data and underlying Fisher curvature, we can construct a learned summary space that asymptotically contains maximal information (Vaart, 1998; Coulton & Wandelt, 2023). We show that this formalism captures relevant information in both a Bayesian inference context as well as for edge aggregation in graphs.

Our approach boasts several advantages. By explicitly learning the score and corresponding inverse-Fisher weights, we are able to construct aggregated summaries that are both asymptotically optimal and robust to changes in data distribution. The result is that we are able to construct optimal summary statistics for independent data for SBI applications, and using the same formalism are able to beat key benchmark GNN learning tasks with far smaller architectures in faster training time than leading networks.

This paper is organised as follows: We first present the notion of information optimality under the likelihood principle and extend to the set-based case. We then present the Fishnets neural embedding method in this framework and review relevant related work in common notation. Next we demonstrate information saturation, robustness, and scalability in a Bayesian context for increasingly difficult problems, and highlight where existing aggregators fall short. We then we show that adopting

Fishnets aggregation as a drop-in replacement for existing GNN architectures allows networks to outperform standard benchmark architectures with fewer learnable parameters and faster training time. Finally, we demonstrate that Fishnets is adept in a stochastic graph-learning scenario by modifying the *ogbn-proteins* dataset.

## 2 METHOD: OPTIMAL AGGREGATION OF INDEPENDENT (HETEROGENEOUS) DATA

### 2.1 FISHER INFORMATION AND OPTIMALITY DEFINITIONS

We first define the notion of information optimality using the likelihood principle. Data $\mathbf{d}$ is related to some parameters (or quantities of interest) via a log-likelihood $\mathcal{L} = \ln p(\mathbf{d}|\boldsymbol{\theta})$. We would like to obtain a compression mapping $f : \mathbf{d} \mapsto \mathbf{t}$ from $N$ data to $n_p$ numbers $\mathbf{t}$ which preserves as much information about the parameters $\boldsymbol{\theta}$ as possible. We define *the information inequality* to quantify how informative this mapping is (Lehmann & Casella, 1998):

$$\mathrm{Var}_{\boldsymbol{\theta}}\left[t_\alpha\right] \geq \left(\mathbf{A}^T\mathbf{F}^{-1}\mathbf{A}\right), \tag{1}$$

where $\mathbf{A} = \nabla\mathbb{E}_{\boldsymbol{\theta}}\left[\mathbf{t}^T\right]$ and the *Fisher Information matrix* is

$$\mathbf{F} = -\mathbb{E}_{\boldsymbol{\theta}}\left[\nabla\nabla^T\mathcal{L}\right] = \mathbb{E}_{\boldsymbol{\theta}}\left[\nabla\mathcal{L}\nabla^T\mathcal{L}\right], \tag{2}$$

where the last equality holds under mild regulatory conditions (Alsing & Wandelt, 2018). The Fisher matrix is the curvature of the log-likelihood, and is in general a function of the parameters. In the case where the compressed numbers $\mathbf{t}$ are unbiased estimators of the parameters, $\mathbb{E}_{\boldsymbol{\theta}}\left[\mathbf{t}\right] = \boldsymbol{\theta}$, $\mathbf{A} = \mathbb{I}$ and the information inequality equation 1 reduces to the Cramér-Rao bound (Cramér, 1946):

$$\mathrm{Var}_{\boldsymbol{\theta}}\left[t_\alpha\right] \geq \mathbf{F}_{\alpha\alpha}^{-1}. \tag{3}$$

Maximising the Fisher information over parameter space decreases the variance on the estimates of the quantities of interest $\boldsymbol{\theta}$. Alsing & Wandelt (2018) show that the score function, $\mathbf{t} = \nabla\mathcal{L}$ saturates the lower bound of equation 1 around a fiducial point, $\boldsymbol{\theta}_*$. We reproduce this proof in the general case over a space of $\boldsymbol{\theta}$ and relate information saturation to parameter estimators in Appendix A.

Maximum likelihood estimators (MLEs) are the asymptotically-optimal estimators for predictive tasks. When they are available, they provide an optimally-informative embedding of the data with respect to the parameters of interest, $\boldsymbol{\theta}$ (see (Alsing & Wandelt, 2018) and Appendix A).

### 2.2 SET-LIKE DATA LIKELIHOODS

Many inference problems consist of a *set* of $n_{\mathrm{data}}$ data vectors, $\{\mathbf{d}_i\}_{i=1}^{n_{\mathrm{data}}}, \mathbf{d}_i \in \mathbb{R}^N$ which obey a global model controlled by parameters $\boldsymbol{\theta} \in \mathbb{R}^{n_p}$, and a possibly arbitrarily deep hierarchy of latent values, $\eta$. The full data likelihood for interesting parameters $\boldsymbol{\theta}$ is given by the integral over latents,

$$p(\{\mathbf{d}_i\}|\boldsymbol{\theta}) = \int p(\{\mathbf{d}_i\}|\boldsymbol{\theta}, \eta)p(\eta|\boldsymbol{\theta})d\eta. \tag{4}$$

When the data are independently distributed, their log-likelihood takes the form

$$\ln p(\{\mathbf{d}_i\}|\boldsymbol{\theta}) = \sum_{i=1}^{n_{\mathrm{data}}} \ln p(\mathbf{d}_i|\boldsymbol{\theta}). \tag{5}$$

A maximum likelihood estimator can then be formed (iteratively) by the Fisher scoring method (Alsing & Wandelt, 2018):

$$\hat{\boldsymbol{\theta}}^{\mathrm{MLE}} = \boldsymbol{\theta}_* + \mathbf{F}^{-1}\mathbf{t}, \tag{6}$$

which requires knowledge of a fiducial point $\boldsymbol{\theta}_*$, the score, $\mathbf{t} \in \mathbb{R}^{n_p}$, and Fisher matrix, $\mathbf{F} \in \mathbb{R}^{n_p \times n_p}$. For problems like linear regression where the analytic form of $\mathbf{F}$ and $\mathbf{t}$ are known, Eq. equation 6 gives the exact MLE for the parameters in a single iteration in the Gaussian approximation, given the dataset. In the case of independent data, both of the score and Fisher information are additive.

Taking the gradient of the log-likelihood with respect to the parameters, the score $\mathbf{t} = \boldsymbol{\nabla_\theta} \ln p(\{\mathbf{d}_i\}|\boldsymbol{\theta})$ for the full dataset is the sum of the scores of the individual data points:

$$\mathbf{t} = \sum_{i=1}^{n_{\text{data}}} \boldsymbol{\nabla_\theta} \ln p(\mathbf{d}_i|\boldsymbol{\theta}) = \sum_{i=1}^{n_{\text{data}}} \mathbf{t}_i(\mathbf{d}_i) \tag{7}$$

Taking the gradient again yields the Hessian, or Fisher information matrix (Amari, 2021; Vaart, 1998) for the dataset,

$$\mathbf{F} = \sum_{i=1}^{n_{\text{data}}} \boldsymbol{\nabla_\theta}\boldsymbol{\nabla_\theta}^T \ln p(\mathbf{d}_i|\boldsymbol{\theta}) = \sum_{i=1}^{n_{\text{data}}} \mathbf{F}_i(\mathbf{d}_i), \tag{8}$$

which is also comprised of a sum of Fisher matrices of individual data. Once the score and Fisher matrix for a dataset are known, the two can be combined to form a pseudo-maximum likelihood estimate (MLE) for the target parameters following equation 6. Therefore, constructing optimal embeddings of independent data with respect to specific quantities of interest just requires aggregating the scores and Fishers, and combining them as in equation 6. However, in general explicit forms for the likelihood (per data vector) may not be known. In this general case, as we will show in the following section, we can parameterize and learn the score and Fisher using neural networks.

## 2.3 TWIN FISHER-SCORE NETWORKS

For many problems, however, the analytic form of the Fisher and score are not known. Here we propose *learning these functions with neural networks*. Due to the additive structure of equation 8 and equation 7, we can parameterize the *per-datapoint* score and Fisher with twin neural networks:

$$\hat{\mathbf{t}}_i = \mathbf{t}(\mathbf{d}_i; w_t) \in \mathbb{R}^{n_p}; \quad \mathbf{t}_{\text{NN}} = \sum_{i}^{n_{\text{data}}} \hat{\mathbf{t}}_i \tag{9}$$

$$\hat{\mathbf{F}}_i = \mathbf{F}(\mathbf{d}_i; w_F) \in \mathbb{R}^{n_p \times n_p}; \quad \mathbf{F}_{\text{NN}} = \sum_{i}^{n_{\text{data}}} \hat{\mathbf{F}}_i \tag{10}$$

where the score and Fisher network are parameterized by weights $w_t$ and $w_F$, respectively. The twin networks output a score and Fisher for each datapoint (see Appendix B for formalism), which are then each summed to obtain a global score and Fisher for the dataset. We can then compute parameter estimates using these aggregated embeddings following equation 6:

$$\hat{\boldsymbol{\theta}}_{\text{NN}}(\{\mathbf{d}_i\}; w_t, w_F) = \mathbf{F}_{\text{NN}}^{-1}\mathbf{t}_{\text{NN}} + c, \tag{11}$$

where the fiducial point $\boldsymbol{\theta}_* = c = 0$ can be set to an arbitrary constant. Provided the embeddings $\hat{\mathbf{t}}_i$ and $\hat{\mathbf{F}}_i$ are learned sufficiently well, the summation formalism can be used to obtain Fisher and score estimates for datasets with heterogeneous structure and arbitrary size. These summaries can be regarded as sufficient statistics, since the score as a function of parameters could in principle be used to reconstruct the likelihood surface up to a constant (Alsing & Wandelt, 2018; Hoffmann & Onnela, 2022).

**Loss Function.** In a regression scenario, we draw data-parameter pairs from the joint distribution $\boldsymbol{\theta}, \{\mathbf{d}_i\} \frown p(\{\mathbf{d}_i\}, \boldsymbol{\theta})$ and compute $\boldsymbol{\theta}_{\text{NN}}$ from equation 11. The twin networks can then be trained jointly using a negative-log Gaussian loss:

$$\mathcal{L}(\boldsymbol{\theta}, \hat{\boldsymbol{\theta}}_{\text{NN}}; w_t, w_F) = \frac{1}{2}(\boldsymbol{\theta} - \hat{\boldsymbol{\theta}}_{\text{NN}})^T \mathbf{F}_{\text{NN}}(\boldsymbol{\theta} - \hat{\boldsymbol{\theta}}_{\text{NN}}) - \frac{1}{2} \ln \det \mathbf{F}_{\text{NN}}. \tag{12}$$

Minimizing this loss with respect to the neural network weights ensures that information is saturated via maximising the aggregated Fisher, and forces the distance between embedding MLE and parameters to be minimized with respect to the Cramér-Rao bound (equation 3) as a function of the data. This loss can also be interpreted as a maximum likelihood (MLE) loss for the quantities of interest $\boldsymbol{\theta}$, as opposed to typical mean-square error (MSE) regression losses (see Appendix E for deepsets details).

## 3 RELATED WORK

**Deepsets Mean Aggregation.** A comparable method for learning over sets of data is regression using the Deepsets (DS) formalism Zaheer et al. (2018). Here an embedding $f(\mathbf{d}_i; w_1)$ is learned for each datum, and then aggregated with a fixed permutation-invariant scheme and fed to a global function $g$; $\hat{\boldsymbol{\theta}} = g\left(\bigoplus_{i=1}^{n_{\text{data}}} f(\mathbf{d}_i; w_1);\ w_2\right)$. The networks are optimised minimising a squared loss against the true parameters, $\text{MSE}(\hat{\boldsymbol{\theta}}, \boldsymbol{\theta})$. When the aggregation is chosen to be the mean, the deepsets formalism is scalable to arbitrary data and becomes equivalent to the Fishnets aggregation formalism *with flat weights across the aggregated data* (see Appendix E for in-depth treatment).

**Learned Softmax Aggregation.** Li et al. present a learnable softmax counterpart to the DS aggregation scheme in the context of edge aggregation in GNNs. Using the above notation, their aggregation scheme reads:

$$\text{SoftmaxAgg}(\cdot) = \sum_{i=1}^{n_{\text{data}}} \frac{\exp\left(\beta f(\mathbf{d}_i; w_1)\right)}{\sum_l \exp\left(\beta f(\mathbf{d}_l; w_1)\right)} \cdot f(\mathbf{d}_i; w_1) \tag{13}$$

where $\beta$ is a learned scalar temperature parameter and $f(\cdot; w_1)$ is some embedding layer. They show that adopting this aggregation scheme allows more graph convolution (GCN) layers to be stacked efficiently to deepen GNN models. Many other aggregation frameworks have been studied, including Graph Attention (Veličković et al., 2018), LSTM units (Hamilton et al., 2017), Recurrent aggregations (Soelch et al., 2019), and scaled multiple aggregators (Corso et al., 2020).

## 4 EXPERIMENTS: BAYESIAN INFORMATION SATURATION

Bayesian Simulation Based Inference (SBI) provides a framework in which to perform inference with intractable likelihood. There have been massive developments in SBI, such as neural ratio estimation (Miller et al., 2021) and density estimation (Alsing et al., 2019; Papamakarios et al., 2019). Key to all of these methods is compressing a large number of data down to small summaries–typically one informative summary per parameter of interest to preserve information (Alsing & Wandelt, 2018; Charnock et al., 2018; Makinen et al., 2021). ML methods like regression (Jeffrey & Wandelt, 2020) and information-maximising neural networks (Charnock et al., 2018; Makinen et al., 2022; 2021) are very good at learning embeddings for highly structured data like images, and can do so losslessly (Makinen et al., 2021). For unstructured datasets comprised of many independent data, the task of constructing optimal summaries amounts to an aggregation task (Zaheer et al., 2018; Hoffmann & Onnela, 2022; Wagstaff et al., 2019). The Fishnets formalism is an optimal version of this aggregation. What deepsets and "learned" aggregation functions are missing is explicitly constructing the inverse-Fisher weights per datapoint, as well as being able to construct the total Fisher information, which is required to turn summaries into unbiased estimators (Alsing & Wandelt, 2018). Explicitly learning the $\mathbf{F}^{-1}$ weights in addition to the score allows us to achieve 1) asymptotic optimality 2) scalability, and 3) robustness to changes in information content among the data.

In this section we demonstrate the 1) information saturation, 2) robustness and 3) scalability of the Fishnets aggregation through two examples in the context of SBI, and highlight the shortcomings of existing aggregators. We first investigate a linear regression scaling problem and introduce a robustness test in which Fishnets outperforms deepset and learned softmax aggregation on test data. We then extend Fishnets to an inference problem with nuisance (latent) parameters and censorship to demonstrate the applicability of network scaling to a regime where MCMC becomes intractable.

### 4.1 VALIDATION CASE: LINEAR REGRESSION

We use a toy linear regression model to validate our method and demonstrate network scalability. We consider the form $y = mx + b + \epsilon$, where $\epsilon \sim \mathcal{N}(0, \sigma)$, where the parameters of interest are the slope and intercept $\boldsymbol{\theta} = (m, b)$. This likelihood has an analytically-calculable score and Fisher matrix (see Appendix C.1), which can be used to calculate exact MLE estimates for the parameters $\boldsymbol{\theta} = (m, b)$ via equation 6. We choose wide Gaussian priors for $\theta$, and uniform distributions for $x \in [0, 10]$ and $\sigma \in [1, 10]$. For network training, we simulate $10^4$ datasets of size $n_{\text{data}} = 500$ datapoints. For testing, we generate an additional $10^4$ datasets of size $n_{\text{data}} = 10^4$ datapoints to demonstrate scalability.

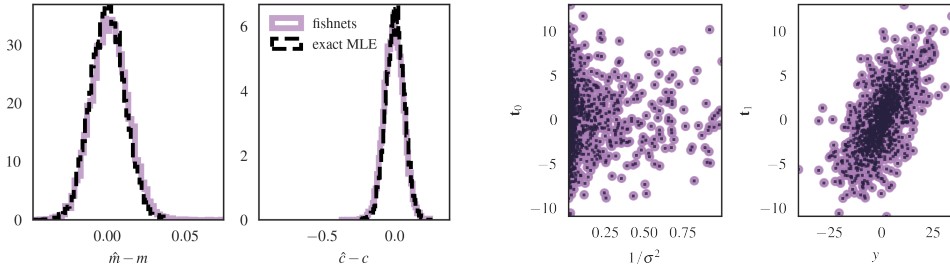

(a) Fishnets saturate information for datasets 20 times larger than the training set.

(b) Fishnets achieve the exact form of the score as a function of input data in the linear regression case.

Figure 1: (a) Residual maximum likelihood estimates for slope (*left*) and intercept (*right*) scatter about the truth for linear regression test datasets of size $n_{\text{data}} = 10^4$. The solid pink line is obtained from a weighted average of an ensemble of Fishnets networks, *which were trained on datasets of size $n_{\text{data}} = 500$*. (b) Slices of true (dark) and network predicted (pink) score vector components as a function of data inputs for the $n_{\text{data}} = 10^4$ test set.

**Results.** We display a comparison of test set performance to the true MLE solution in Figure 1(a), and slices of the true and predicted score vectors as a function of input data. The networks are able to recover the exact score and Fisher information matrices (see Figure 1(b)), even when scaled up 20-fold. *This test demonstrates that Fishnets can (1) saturate information on small training sets to enable scalable predictions on far larger aggregations of data (2).*

### 4.2 ROBUSTNESS TO CHANGES IN THE UNDERLYING DATA DISTRIBUTIONS

In real-world applications, actual data processed by a network might follow a different distribution than that seen in training. Here we compare three different network formalisms on changing shapes of target data distributions.

We train three networks on the same $n_{\text{data}} = 500$ datasets as before: a sum-aggregated Fishnets network, a mean-aggregated deepset, and a learned softmax-aggregated deepset with mean-square error loss. To demonstrate the improvement in aggregation Fishnets offers, we adopt smaller networks for the regression task (see Table 1 and Appendix C.2 for architecture details). We apply our

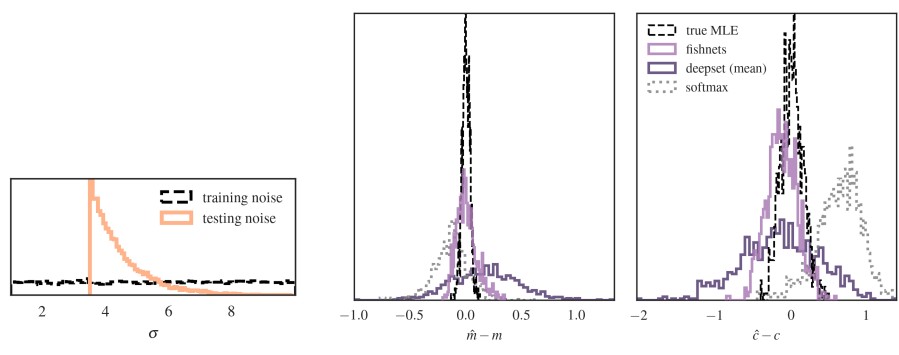

(a) Changing noise distribution.

(b) Robustness testing different aggregators

Figure 2: (b) Fishnets (pink) are robust to different noise distributions in test data (2(a)). Deepsets (grey) can return biased results for some parameters (left) and lossy estimates for others (right). Learned softmax aggregation appears to provide lossier and biased parameter estimates.

trained networks to test data $n_{\text{data}} = 850$ with noise variances and $x$ values drawn from different distributions to the training data: $\sigma \curvearrowright \text{Exp}(\lambda = 1.0)$ centred at $\sigma = 3.5$, truncated at $\sigma = 10.0$,

|  | network | # params | MSE$(\hat{m}, m_{\text{true}})$ | MSE$(\hat{c}, c_{\text{true}})$ |
|---|---|---|---|---|
| robustness test | **fishnets** | $10,855$ | **$0.007 \pm 0.017$** | **$0.046 \pm 0.078$** |
|  | deepset | $87,810$ | $0.120 \pm 0.178$ | $0.285 \pm 0.406$ |
|  | softmax | $87,811$ | $0.042 \pm 0.069$ | $0.482 \pm 0.347$ |

Table 1: Summary of robustness testing for different set-based networks. Fishnets' Fisher aggregation has an advantage over mean- and learned softmax deepsets aggregation when test data follows a different distribution than the training suite, and does so with an eigth of the number of learnable parameters.

and $x \curvearrowleft \mathcal{U}(0, 3)$. The noise and covariate distributions have the same support as the training data, but have different expectation values and distributions, which can pose a problem for the mean-aggregation used in the standard deepsets formalism. We display results in Figure 2(b). The heterogeneous Fishnets aggregation allows the network to correctly embed the noisy data drawn from the different distributions, while a significant loss in information can be seen for flat mean aggregation. The learned softmax aggregation improves the width of the residual distribution, but is still significantly wider than the Fishnets solution. We quote numeric results in Table 1.

These robustness tests show that Fishnets successfully learns *per-object* embeddings (score) and weights (Fisher) within sets, *while being robust to changing shapes of the training distributions of these quantities (3)*. This test also shows that even in a very simple prediction scenario, *common and learned aggregators can suffer from robustness issues.*

### 4.3 SCALABLE INFERENCE WITH CENSORSHIP AND NUISANCE PARAMETERS

As a non-trivial information saturation example we consider a censorship inference problem with latent parameters inspired by epidemiological studies. Consider a serum which, when injected into a patient, decays over time, and the (heterogeneous) decay rate among people is not well known. A population of patients are injected with the serum and then asked to come back to the lab within $t_{\text{max}} = 10$ days for a measurement of the remaining serum-levels in their blood, $s$. We can cast this problem as a Bayesian hierarchical model visualised in Figure 3(a) (see Appendix C.3 for details) where the goal is to infer the mean $\mu$ and scale $\Theta$ of the decay rate Gamma distribution from the data, $\{\tau_i, s_i\}$. In the censored case, measurements are rejected if $s_i < s_{\text{min}}$, and collected until $n_{\text{data}}$ valid samples are collected. As a ground-truth comparison for the uncensored version of this problem, we sample the above hierarchical model using Hamiltonian Monte-Carlo (HMC). For comparison, we utilize the same Fishnets architecture and small-data training setup as before to predict $(\mu, \Theta)$ from data inputs $[\tau_i, s_i]^T$. Once trained, we generated a new suite of $n_{\text{data}} = 500$ simulations and pass the data through the network to learn a neural posterior from $(\hat{\theta}_{\text{NN}}, \boldsymbol{\theta})$ pairs. We then evaluated both HMC and neural posteriors at the same target data. Finally, using the same network we perform the same procedure, this time with simulations of size $n_{\text{data}} = 10^4$, *where the HMC becomes computationally prohibitive.*

**Results.** We display inference results in Figure 3(b). The summaries obtained from Fishnet compression of the small data (green) result in posteriors that hug the "true" MCMC contours (black), *indicating information saturation*. Extending the same network on the larger data results in intuitively smaller contours (blue). It should be emphasized that $n_{\text{data}} = 10^4$ *is a regime where the MCMC inference is no longer tractable on standard devices.* Fishnets here allows for 1) much faster posterior calculation and 2) allows for optimal inference on larger data set sizes without any retraining.

As a final demonstration we solve the same problem, this time subject to censorship. In the censored case, the target joint posterior defined by the hierarchical model requires computing an integral for the selection probability as a function of the model hyper-parameters; in general, these selection terms make Bayesian problems with censorship computationally challenging, or intractable in some cases (Qi et al., 2022; Dickey et al., 1987).

We train Fishnets on the small data size, subject to censorship below $s_{\text{min}}$. We obtain posteriors of the same shape of the censored case, but for a consistency check perform a probability-integral transform (PIT) test for the neural posterior. For each parameter we want the marginal PIT test to yield a uniform distribution to show that the learned posterior behaves as a continuous distribution. We display these results in Figure 5. We obtain a Kolmogorov-Smirnov test (Massey Jr., 1951)

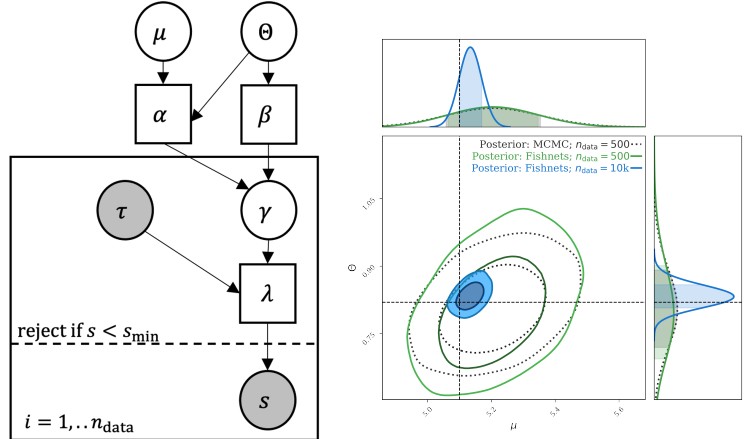

(a) Gamma population hierarchical Bayesian model diagram.

(b) Information Saturation against MCMC.

Figure 3: (a) Gamma population plate diagram. Circles represent random variables, boxes are deterministic quantities, and shaded variables are observed as data. The dashed line represents a possible censorship in measurement. Measurements of data $(t, s)_i$ are conducted until $n_{\text{data}}$ samples are drawn. (b) The same Fishnets network can be used for inference on datasets much larger than those used in training. The twin Fishnet architecture was trained on $n_{\text{data}} = 500$. We then compress a target dataset and perform density estimation (green) and compare to an MCMC sampler as our true posterior (black dashed). Fishnets nearly saturates the information. We then *use the same network to compress simulations of* $n_{\text{data}} = 10^4$ to obtain the blue contours.

p-value of 0.628 and 0.233 for parameters $\mu$ and $\Theta$, respectively, indicating that our posterior is well-parameterized and robust.

## 5 GRAPH NEURAL NETWORK AGGREGATION

Graphs can be thought of as tuples of sets within connected neighborhoods. Graph neural networks (GNNs) operate by message-passing along edges between nodes. For predicting node- and graph-level properties, an aggregation of these sets of edges $\{\mathbf{e}_{ij}\}$ or nodes $\{\mathbf{v}_i\}$ is required to reduce features to fixed-size feature vectors. Whereas in the SBI setting, we are interested in finding optimal estimators for specific parameters of interest, in the GNN aggregation setting we are implicitly trying to find a compact latent (embedding) representation of the aggregated neighborhood data, and optimally estimate and propagate those latent features through the GNN architecture.

Here we compare the Fishnets aggregation scheme as a drop-in replacement for learned softmax aggregators within the graph convolutional network (GCN) scheme presented by Li et al.. We can rewrite our aggregations to occur within neighborhoods of nodes:

$$\text{SoftmaxAgg}(\cdot) = \sum_{i \in \mathcal{N}(v)} \frac{\exp\left(\beta \mathbf{e}_{iv}\right)}{\sum_{l \in \mathcal{N}} \exp\left(\beta \mathbf{e}_{li}\right)} \cdot \mathbf{e}_{iv}, \tag{14}$$

$$\text{FishnetsAgg}(\cdot) = \left(\sum_{i \in \mathcal{N}(v)} \mathbf{F}(\mathbf{e}_{iv})\right)^{-1} \left(\sum_{i \in \mathcal{N}(v)} \mathbf{t}(\mathbf{e}_{iv})\right), \tag{15}$$

where the aggregation occurs in a neighborhood $\mathcal{N}$ of a node $v$. The Fishnets aggregation requires a bottleneck hyperparameter, $n_p$, which controls the size of the score embedding $\mathbf{t}(\mathbf{e}_{iv}) \in \mathbb{R}^{n_p}$ and Fisher Cholesky factors $\mathbf{F}_{\text{chol}} \in \mathbb{R}^{n_p(n_p+1)/2}$. We use a single linear layer before aggregation to obtain score and Fisher components from hidden layer embeddings.

## 5.1 DROP-IN REPLACEMENT FOR GRAPH BENCHMARK DATASETS

Here we replace the learned softmax aggregation with Fishnets aggregation in Li et al.'s publicly-available best-performing models. We change four hyperparameters in testing our new architectures: number of layers, $n_p$, dropout, and learning rate. We study several graph datasets from the Open Graph Benchmark (OGB) (Hu et al., 2020; 2021), which require substantial aggregation steps to predict either node or graph-level properties. The object of this study is to investigate *how well fishnets aggregation can perform within an existing architecture*, with fewer layers and minimal hyperoptimisation.

| ogb dataset performance | | | | ogb-proteins comparison | | |
|---|---|---|---|---|---|---|
| dataset | GCN | **fishnets** | | model | # params | test ROC-AUC |
| ogb-arxiv | 0.7100 | 0.7062 | | GCN-112 | 1,887,144 | $0.8425 \pm 0.0018$ |
| ogb-molhiv | 0.7600 | **0.8000** | | fishnets-8 | 146,596 | $0.8410 \pm 0.0013$ |
| ogb-proteins | 0.8425 | **0.8444** | | **fishnets-16** | 280,740 | $\mathbf{0.8444 \pm 0.0018}$ |

Table 2: (*left*) Summary of benchmark improvement within GCN framework with Fishnets aggregation. (*right*) Model size comparison for ogb-proteins benchmark. Fishnets aggregation improves performance with $\sim 15\%$ of the learnable parameters.

**Results.** We display benchmark results in Table 2, and refer the reader to Appendix D for architecture and dataset details. This small drop-in study shows that incorporating the more information-efficient Fishnets Aggregation, we can achieve better than or similar results to SOTA GCNs *with a fraction of the trainable parameters and training epochs*.

## 5.2 FOCUS STUDY ON *ogbn-proteins* BENCHMARK

In this section we study the proteins dataset in detail highlight a scenario where the heterogeneous Fishnets aggregation drastically improves performance. Here we expect different node neighborhoods to have a heterogeneous edge weighting "association score" structure across protein categories, making the Fishnets aggregation ideal for applicability beyond the training set, as in the linear regression case. The association scores can be stochastically modelled with added measurement noise, increasing the difficulty of the classification problem. We adopt a stripped-down version of the training routine presented in Li et al. (2020) (no subgraph and edge preprocessing) to make modifications to the raw data by adding noise. We first benchmark our training routine with smaller GCN and Fishnets aggregation on the noise-free data, and then proceed to adding noise to the edges.

**Noisefree Results.** We display representative test ROC-AUC curves over training in Figure 4(a), and in Table 3. Fishnets-16 and Fishnets-20 clearly saturate information within 250 epochs to 79.63% and 81.10% accuracy respectively.

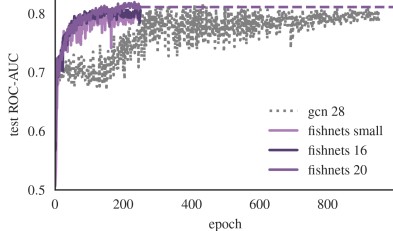 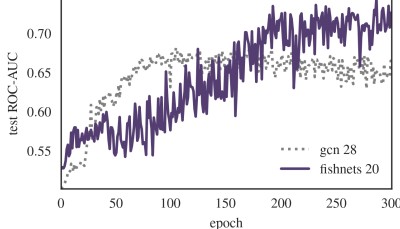

(a) Aggregation test performance on ogbn-proteins. Fishnets saturates the patience criteria within 250 epochs (dashed line).

(b) Test performance in noisy edge setting.

Figure 4: Representative Test ROC-AUC curves for (a) benchmark and (b) noisy proteins datasets. Fishnets aggregation within GCNs clearly saturates information more quickly than GCNs and can also handle noisy edges and contextual information through explicit weight parameterization (b).

| test | network | # params | test ROC-AUC |
|---|---|---|---|
| noisefree | **fishnets-20** | $442,372$ | **$0.8110 \pm 0.0021$** |
| | GCN-28 | $477,964$ | $0.7951 \pm 0.0059$ |
| noisy edges | **fishnets-20** | $442,500$ | **$0.7198 \pm 0.0109$** |
| | GCN-28 | $478,092$ | $0.6471 \pm 0.0090$ |

Table 3: Summary of performance on benchmark and noisy variants of the proteins dataset. Errorbars denote standard deviation of test ROC-AUC in the last ten epochs of training.

### 5.2.1 MODELLING UNCERTAIN PROTEIN ASSOCIATIONS.

Here we incorporate uncertainties on the protein interaction strengths (edges), in order to demonstrate the robustness of the Fishnets approach to changes in the underlying data (noise) distribution on the graph features. We model noisy "measurements" of the protein graph edge associations using a simple Binomial model: taking the dataset edges $\mathbf{p}_{ij} = \mathbf{e}_{ij} \in [\mathbf{0}, \mathbf{1})$ as the "true" association strengths, we can simulate a noisy measurement of those quantities as $N$ weighted coin tosses per edge, where $N$ varies between measurements:

$$N \curvearrowleft \mathcal{U}(20, 200) \tag{16}$$

$$\mathbf{n}_{\text{success}} \curvearrowleft \text{Binomial}\left(n = N, p = \mathbf{p}_{ij}\right); \ \mathbf{e}_{ij} \leftarrow \left[\hat{\mathbf{p}}_{ij} = \mathbf{n}_{\text{success}}/N, N\right]. \tag{17}$$

Note that in the last step the new graph edge now contains the (noisy) measured associations, as well as $N$ (which provides a measure of uncertainty on those estimated interaction strengths). The GNN task is now to learn to re-weight predictions conditioned on the provided $N$ coin toss information, much like feeding in $\sigma$ in the linear regression case. We train a 28-layer GCN and 20-layer Fishnets. For the test dataset, we alter the distribution for $N$ to be $\mathcal{U}(20, 50) + \mathcal{U}(170, 200)$ such that we sample the extremes of the training distribution support.

**Noisy Results.** We display test ROC-AUC curves for both networks in Figure 4(b), subject to a patience setting of 250 epochs on the validation set. The GCN framework exhibits an early plateau at $64.71\%$ accuracy, while Fishnets saturates to $71.98\%$ accuracy. This stark difference in behaviour can be explained by the difference in formalism: The Fishnets aggregation explicitly learns a weighting scheme as a function of measured edge probabilities *and* the conditional information $N$, much like the linear regression case where $\sigma$ was passed as an input. This scheme helps to learn how to deal with edge-case noise artefacts like the noisy edge test case. Explicitly specifying the inverse-Fisher weighting formalism as an inductive bias (Battaglia et al., 2018) during aggregation can help explain the fast information saturation exhibited in both graph test settings.

## 6 DISCUSSION & FUTURE WORK

In this paper we built up an information-theoretic approach to optimal aggregation in the form of Fishnets. Through progressively non-trivial examples, we demonstrated that explicitly parameterizing the score and inverse-Fisher weights of set members results in an aggregation scheme that saturates Bayesian information in non-trivial problems, and also serves as an optimal aggregator for graph neural networks.

The stark improvement in information saturation on the proteins test dataset relative to architecture size and training efficiency indicates that the Fishnets aggregation acts as an information-level inductive bias for GNN aggregation. Follow-up study is warranted on optimizing hyperparameter choices for graph neural network architectures using Fishnets. We chose to demonstrate improved information capture by using an ablation study of smaller models, but careful (and potentially bigger) network design would almost certainly improve results here and potentially achieve SOTA accuracy on common benchmarks.

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

## A    SATURATING THE INFORMATION INEQUALITY OVER PARAMETER SPACE

Here we show that knowing the score function $\mathbf{t} = \nabla\mathcal{L}$ saturates the information inequality and provides a natural data compression function. We first consider the Taylor expansion of the log-likelihood around a fixed fiducial point in parameter space, $\boldsymbol{\theta}_*$, (where $g_* \equiv g(\boldsymbol{\theta} = \boldsymbol{\theta}_*)$):

$$\mathcal{L} = \mathcal{L}_* + \delta\boldsymbol{\theta}^T \nabla\mathcal{L}_* - \frac{1}{2}\delta\boldsymbol{\theta}^T \mathbf{J}_* \delta\boldsymbol{\theta} \tag{18}$$

where $\mathbf{J} = -\nabla\nabla^T\mathcal{L}$ is the observed information matrix. To linear order in $\boldsymbol{\theta}$, the data $\mathbf{d}$ couples to the parameters through the score function $\mathbf{t} \in \mathbb{R}^{n_p}$. We can show that $\mathbf{t}$ saturates the information inequality via

$$\mathrm{Cov}_{\boldsymbol{\theta}}\left[\mathbf{t}, \mathbf{t}\right] = \mathbb{E}_{\boldsymbol{\theta}}\left[\nabla\mathcal{L}_* \nabla_*^T\right] = \mathbf{F}_*, \tag{19}$$

where we have used the fact that $\mathbb{E}_{\boldsymbol{\theta}}\left[\nabla\mathcal{L}_*\right] = 0$. From this we observe that the covariance of the score function is the Fisher matrix. Using the fact that

$$\mathbf{A} = \nabla\mathbb{E}_{\boldsymbol{\theta}}\left[\nabla^T\mathcal{L}\right] = \mathbb{E}_{\boldsymbol{\theta}}\left[\nabla\nabla^T\mathcal{L}\right] = -\mathbf{F}_*, \tag{20}$$

the right-hand side of the information inequality becomes $\mathbf{A}_*^T \mathbf{F}_*^{-1} \mathbf{A}_* = \mathbf{F}_*$, which shows that the score statistics $\mathbf{t}$ saturate the information inequality. Within this formalism, no statistics can provide more (Fisher) information about the parameters $\boldsymbol{\theta}$.

We can relate this information saturation to an optima, quasi maximum-likelihood estimator whose covariance is equal to the inverse Fisher information from the above derivation. Maximising the Taylor expansion equation 18 with respect to the parameters yields

$$\hat{\boldsymbol{\theta}} = \boldsymbol{\theta}_* + \mathbf{J}_*^{-1}\nabla\mathcal{L}_* \tag{21}$$

where both the score $\mathbf{t}_* = \nabla\mathcal{L}_*$ and the observed information $\mathbf{J}_*^{-1}$ depend on the observed data. In practice, we can exchange $\mathbf{J}$ with its expectation value, the Fisher information: $\mathbf{F}_* \equiv \mathbb{E}_{\boldsymbol{\theta}}\left[\mathbf{J}_*\right]$, which yields

$$\hat{\boldsymbol{\theta}} = \boldsymbol{\theta}_* + \mathbf{F}_*^{-1}\nabla\mathcal{L}_*. \tag{22}$$

Making this replacement means the MLE estimator only depends on the data through the score function statistics $\mathbf{t} = \nabla\mathcal{L}_*$. The covariance of the MLE estimator (at the expansion point $\boldsymbol{\theta}_*$) is then:

$$\mathrm{Cov}_{\boldsymbol{\theta}_*}\left[\hat{\boldsymbol{\theta}}, \hat{\boldsymbol{\theta}}\right] = \mathbf{F}_*^{-1}\mathbb{E}_{\boldsymbol{\theta}_*}\left[\nabla\mathcal{L}_* \nabla^T\mathcal{L}_*\right] \mathbf{F}_*^{-1} = \mathbf{F}_*^{-1}, \tag{23}$$

where $\mathbb{E}_{\boldsymbol{\theta}_*}\left[\nabla\mathcal{L}_* \nabla^T\mathcal{L}_*\right] \equiv \mathbf{F}_*$. Hence the covariance of the MLE is equal to the Fisher information matrix at $\boldsymbol{\theta}_*$ and the Cramér-Rao bound is saturated.

The above proof of information saturation calculated the information saturation around a fixed fiducial point. In general, however, by parameterising the score and Fisher functions with neural networks under the loss equation 12 we are learning an embedding $\mathbf{t}(\mathbf{d}_i; w_t)$ and weighting neighborhood $\mathbf{F}(\mathbf{d}_i; w_F)$ as a function of data (and implicitly parameters).

## B    CALCULATING THE FISHER MATRIX FROM NETWORK OUTPUTS

To ensure that our Fisher matrix is positive-definite, our Fisher-score networks output $n_{\mathrm{params}} + n_{\mathrm{params}}\frac{(n_{\mathrm{params}}+1)}{2}$ numbers as lower triangular entries in a Cholesky decomposition of the Fisher matrix, $\mathbf{L}$. To ensure that the lower triangular entries remain positive-definite, we add a softplus activation to the diagonal entries of $\mathbf{L}$:

$$\mathrm{diag}(\mathbf{L}) \leftarrow \mathrm{softplus}(\mathrm{diag}(\mathbf{L}) \tag{24}$$

We then compute the Fisher via:

$$\mathbf{F} = \mathbf{L}\mathbf{L}^T \tag{25}$$

The negative-log likelihood loss in Equation 12 allows for explicit interrogation of the resulting Fisher matrix at the level of the predicted quantities (parameters), and ensures that the summary space in $\hat{\theta}$ is convex. In the GNN regression formalism, Fishnets does not *explicitly* maximise the Fisher information as a part of the loss, rather the Fisher matrix weights are optimized as an inductive bias as a hidden layer in the GNN scheme. E

## C  BAYESIAN INFORMATION EXPERIMENT DETAILS

### C.1  SCALABLE LINEAR REGRESSION

We use a toy linear regression model to validate our method and demonstrate network scalability. We consider the form $y = mx + b + \epsilon$, where $\epsilon \sim \mathcal{N}(0, \sigma)$, where the parameters of interest are the slope and intercept $\boldsymbol{\theta} = (m, b)$. This likelihood has an analytically-calculable score and Fisher matrix,

$$\mathbf{t} = \sum_{i=1}^{n_{\text{data}}} \frac{1}{\sigma_i^2} \begin{bmatrix} x_i(y_i - (m_{\text{fid}}x_i + b_{\text{fid}})) \\ y_i - (m_{\text{fid}}x_i + b_{\text{fid}})) \end{bmatrix} + \mathbf{t}_0, \tag{26}$$

$$\mathbf{F} = \sum_{i=1}^{n_{\text{data}}} \frac{1}{\sigma_i^2} \begin{bmatrix} x_i^2 & x_i \\ x_i & 1 \end{bmatrix} + \mathbf{C}_{\text{p}}^{-1}, \tag{27}$$

where $\mathbf{t}_0 = \mathbf{C}_{\text{p}}^{-1}(\theta_{\text{fid}} - \mu_{\text{p}})$, with $\mu_{\text{p}} = \mathbf{0}$ is the mean of the prior on the score, and $\mathbf{C}_{\text{p}}^{-1} = \mathbf{I}$ is added to the Fisher matrix as a prior on the inverse-covariance of the spread of the summaries. With these two expressions we can calculate exact MLE estimates for the parameters $\boldsymbol{\theta} = (m, b)$ via Eq. equation 6. We choose wide Gaussian priors for $\theta$, and uniform priors for $x \in [0, 10]$ and $\sigma \in [1, 10]$. For network training, we simulate $10^4$ datasets of size $n_{\text{data}} = 500$ datapoints. For testing, we generate an additional $10^4$ datasets of size $n_{\text{data}} = 10^4$ datapoints to demonstrate scalability. We use fully-connected MLPs of size [256, 256, 256] with `ELU` activations (Clevert et al., 2015) for both score and Fisher networks. Both networks receive the input data $[y_i, x_i, \sigma_i^2]^T$. We train networks for 2500 epochs with an `adam` optimizer using a step learning rate decay schedule. We train an ensemble of 10 networks in parallel on the same training data with different initializations.

### C.2  ROBUSTNESS NETWORK ARCHITECTURE COMPARISON

We train three networks on the same $n_{\text{data}} = 500$ datasets as before: a sum-aggregated Fishnets network, a mean-aggregated deepset, and a learned softmax-aggregated deepset (no Fisher output and standard MSE loss against true parameters $\text{MSE}(\hat{\boldsymbol{\theta}}, \boldsymbol{\theta})$). Here we initialise Fishnets with [50,50,50] hidden units for score and Fisher networks, and two embeddings of [128,128,128] hidden units for both deepset networks, all with `swish` (Ramachandran et al., 2017) nonlinearities for the data embedding (see Table 1). All networks are initialised with the same seed (see Appendix C.2 for architecture details).

### C.3  GAMMA POPULATION MODEL

Consider a serum which increases patients' red blood cell counts, whose decay rate, $\tau$, is not known. A population of patients are injected with the serum and then asked to come back to the lab within $t_{\text{max}} = 10$ days for a measurement of their blood cell count, $s$. We can cast this problem using the following hierarchical model

$$\mu \curvearrowright \mathcal{U}(0.5, 10)$$
$$\Theta \curvearrowright \mathcal{U}(0.1, 1.5)$$
$$\gamma_i \curvearrowright \text{Gamma}(\alpha = \mu/\Theta, \beta = 1/\Theta)$$
$$\tau_i \curvearrowright \mathcal{U}(0, 10)$$
$$\lambda_i = A \exp(-\tau_i/\gamma_i)$$
$$s_i \curvearrowright \text{Pois}(\lambda_i),$$

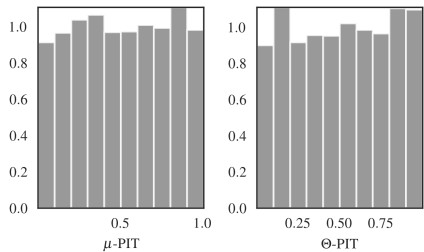

Figure 5: Density estimation posteriors obtained from parameter-Fishnets summary pairs are robust over training data. Each parameter's PIT test is close to uniform, which shows that the Fishnets summary posterior has successfully captured the underlying Bayesian information from the data.

where the goal is to infer the mean $\mu$ and scale $\Theta$ of the decay rate Gamma distribution from the data, $\{\tau_i, s_i\}$. In the censored case, measurements are rejected if $s_i < s_{\min}$, and collected until $n_{\text{data}}$ samples are accepted. The model is visualised in a plate diagram in Figure 3(a). In the uncensored case, the posterior estimation for this problem is readily solved using a high-dimensional Hamiltonian Monte-Carlo (HMC) sampler. We implement this model in `Numpyro` (Phan et al., 2019) as a baseline MCMC comparison for our algorithm. For the Fishnets implementation, we generate $10^4$ simulations of size $n_{\text{data}} = 500$ over a uniform prior for $\mu$ and $\Theta$. We then train the same Fishnets architecture used for the linear regression case with data inputs $[\tau_i, s_i]^T$. Once the networks were trained, we pass a suite of $n_{\text{data}} = 5000$ simulations through the network to generate neural summaries with which to train a density estimation network. Following Alsing et al. (2019), we use Mixture Density Networks to learn an amortized posterior for $p(\hat{\theta}_{\text{NN}}|\theta)$ with three hidden layers of size $[50, 50, 50]$. We then evaluate this posterior at the same target data used for the HMC, shown in green in Figure 3(b). The Fishnets compression results in slightly inflated contours, indicating a small leakage of information. To demonstrate scaling, we additionally generate another simulation at $n_{\text{data}} = 10^4$ using the same random seed. We train another amortised posterior using 5000 simulations at $n_{\text{data}} = 10^4$ and pass the data through the same trained Fishnet architecture. The resulting posterior is shown in blue for comparison.

## D  GRAPH PREDICTION BENCHMARK EXPERIMENT DETAILS

All GNN models are implemented in PyTorch Geometric (Fey & Lenssen, 2019), and all experiments are performed on a single NVIDIA V100 32GB with the same random seed for initialisation and training. We first describe graph- and node-level prediction tasks, followed by the experimental details on the three benchmark datasets.

**Node Property Prediction.** This task consists of aggregating edge and node information within neighborhoods to predict properties at the node level. The *ogbn-arxiv* dataset is a directed citation graph of $169,343$ papers summarised as 128-dimensional vectors (nodes) and $1,166,243$ citations (edges), where the direction indicates the citation direction. The task is to predict which of 40 classes each paper belongs to. The *ogbn-proteins* dataset consists of $132,534$ proteins encoded as 8-dimensional one-hot features indicating protein species (nodes) and $39,561,252$ undirected weighted edges indicating association scores between proteins. The task a 112-class classification from aggregated subgraph edges and nodes using an ROC-AUC metric.

**Graph Property Prediction.** This task requires the aggregation of edges and nodes to global features of a graph. We consider the *ogbn-molhiv* dataset, which is comprised of $41,127$ molecules with atoms arranged as nodes and bonds as edges. The prediction task is binary classification.

**ogb-molhiv.** This dataset does not provide a node feature for each protein. We initialize the node features via a sum aggregation, e.g. $x_i = \sum_{j \in \mathcal{N}} e_{ij}$, where $x_i$ denotes the initialized node features and $e_{ij}$ denotes the input edge features. We train a 7-layer DyResGEN model with softmax aggregator with learnable $\beta$ parameter. A batch normalization is used for each layer. We set the hidden channel size as 256. A dropout with a rate of 0.5 is used for each layer. An Adam optimizer with a learning rate of 0.0001 are used to train the model for 150 epochs. For the Fishnets comparison we train the

same 7-layer network with Fishnets Aggregation with bottleneck $n_p = 8$ in place of the softmax aggregation, and a learning rate of 0.00002.

**ogb-arxiv.** We train Li et al. (2020)'s 28-layer ResGEN model with softmax aggregation where $\beta$ is fixed as 0.1. Full batch training and test are applied. A batch normalization is used for each layer. The hidden channel size is 128. We apply a dropout with a rate of 0.5 for each layer. An Adam optimizer with a learning rate of 0.01 is used to train the model for 500 epochs. For the Fishnets comparison we train a 3-layer version of the same ResGEN network with Fishnets Aggregation with bottleneck $n_p = 9$ in place of the softmax aggregation.

**ogb-proteins.** This dataset does not provide a node feature for each protein. We initialize the node features via a sum aggregation, e.g. $x_i = \sum_{j \in \mathcal{N}} e_{ij}$ , where $x_i$ denotes the initialized node features and $e_{ij}$ denotes the input edge features. We train Li et al. (2020)'s 112-layer DyResGEN with softmax aggregator. A hidden channel size of 64 is used. A layer normalization and a dropout with a rate of 0.1 are used for each layer. We train the model for 1000 epochs with an Adam optimizer with a learning rate of 0.01. For the Fishnets comparison we train an 8-layer and 16-layer version of the same DyResGEN network with Fishnets Aggregation with bottleneck $n_p = 8$ in place of the softmax aggregation. Here we temper the learning rate to 0.005 and decrease the dropout rate to 0.25.

### D.1 NOISY PROTEINS FOCUS STUDY

Here we again initialize the node features via a sum aggregation.

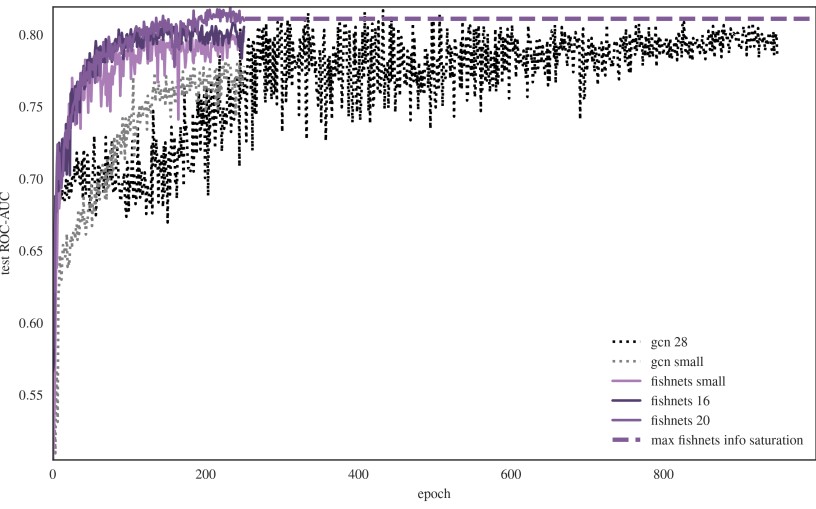

Figure 6: Zoomed-in test ROC-AUC training trajectories for models considered in benchmark ablation study on ogbn-proteins.

We test five model architectures using the vanilla *ogbn-proteins* dataset (no subgraph and edge preprocessing as performed by Li et al. (2020)). This change allowed us to flexibly incorporate the added edge feature in the noisy edge setting. To benchmark our training routine we adopt a 28-layer DyResGEN network with learned softmax aggregations and hidden size of 64, and a smaller version of this model with hidden size 14 and 28 layers. We construct two, shallower Fishnets GNNs, with 16 and 20 layers, each with 64 hidden units, and one small model with 14 hidden units and 14 layers. For each graph convolution aggregation, we adopt a "score" bottleneck of $n_p = 10$ for the large Fishnets models and $n_p = 8$ for the small model. We train all networks with a cross-entropy loss over the same dataset and fixed random seed using an Adam optimizer with fixed learning rate $0.001$. We incorporate an early stopping criterion conditioned on the validation data, which dictates an end to training (saturation) when the validation ROC-AUC metric stops increasing for `patience = 250` epochs.

In the noisy proteins setting we again control for stochasticity in training set loading and added edge noise by fixing the initial random seed before each training run.

| test | network | # params | test ROC-AUC |
|---|---|---|---|
| noisefree | **fishnets-20** | $442,372$ | **$0.8110 \pm 0.0021$** |
| | fishnets-16 | $355,584$ | $0.7963 \pm 0.0059$ |
| | fishnets small | $30,360$ | $0.7929 \pm 0.0045$ |
| | GCN-28 | $477,964$ | $0.7951 \pm 0.0059$ |
| | GCN small | $33,580$ | $0.7731 \pm 0.0052$ |
| noisy edges | **fishnets-20** | $442,500$ | **$0.7198 \pm 0.0109$** |
| | GCN-28 | $478,092$ | $0.6471 \pm 0.0090$ |

Table 4: Full summary of performance on benchmark and noisy variants of the proteins dataset. Errorbars denote standard deviation of test ROC-AUC in the last ten epochs of training.

## E   DEEPSETS FORMALISM

**Summary.** The deepsets method presented by Zaheer et al. shows in Theorem 9 that any function over a *countable* set can be decomposed in the form $f(X) = \rho\left(\sum_{x \in X} \phi(x)\right)$. They then extend this to the universality of deepsets since $\rho$ and $\phi$ can be parameterized as neural networks, which can be universal function approximators. The deepsets formalism allows point-estimates for regression parameters to be obtained following an aggregation of features in a potentially variably-sized set of data. Incorporating our formalism, each set member $\mathbf{d}_i$ is first passed to a neural network $f(\mathbf{d}_i; w_1)$, and subsequently aggregated using some permutation-invariant scheme, $\bigoplus_i$.

$$\hat{\boldsymbol{\theta}} = g\left(\bigoplus_{i=1}^{n_{\text{data}}} f(\mathbf{d}_i; w_1);\ w_2\right), \tag{28}$$

where $f$ is the embedding network and $g$ is the "global" function that maps aggregated features to predicted parameters. When the aggregation is chosen to be the mean, the deepsets formalism is scalable to arbitrary data and becomes equivalent to the Fishnets aggregation formalism *with flat weights across the aggregated data*. The loss takes the form of a convex squared loss, e.g. the mean square error

$$\mathcal{L} = \frac{1}{n_{\text{batch}}} \sum_{i}^{n_{\text{batch}}} (\hat{\boldsymbol{\theta}}_i - \boldsymbol{\theta}_i)^2 \tag{29}$$

where $n_{\text{batch}}$ is a batch of full simulations, each of size $n_{\text{data}}$.

**Training and Generalization.** In practice, Deepsets requires a *fixed* aggregation scheme from which to learn its global function. Most often this is a summary of embedding layers $\phi(x)$. For networks to scale to arbitrary dataset cardinality, aggregations like max, mean, and variance need to be used. In a scenario where the training data distribution $p(x, \theta)$ follows a different distribution from the training data, these aggregations might pose an issue. Concretely, consider $x \sim \mathcal{N}(\mu, 1)$, with the target quantity $\theta = \mu \sim \mathcal{U}(0, 2)$. Next consider a deepset with the identity embedding layer $\phi(x) = x$ and mean-aggregation:

$$\hat{\mu} = \rho\left(\frac{1}{n} \sum_i x_i\right) \tag{30}$$

If test data $x_i$ were drawn from the same distribution as the test data, $\rho$ would act on the mean value of the set of data, in this case $\rho(\mathbb{E}_{p(x,\theta)}[x_i])$, and would converge to a *learned function of the joint prior-data distribution* $p(x, \theta)$. However, if a test set of data were drawn from a different distribution, e.g. $\mu_{\text{test}} \sim \mathcal{N}(0.5, 0.1)$, then the expectation $\mathbb{E}_{p(x,\theta)}$ would take on a different value, and $\rho$ would return an incorrect result for the deterministic aggregation. Here it is important to emphasize that $p^{\text{test}}(x, \theta)$ and $p(x, \theta)$ overlap along the same support, meaning the network *will have seen examples of data drawn from this prior* in the limit of an infinite training set. However, the fixed aggregation makes use of a training-data distribution-dependent quantity for its mapping, which can be skewed under covariate shift or different noise settings.

