# OpenReview forum: "Fishnets: Information-Optimal, Scalable Aggregation for Sets and Graphs"
_ICLR.cc/2024/Conference — Submitted to ICLR 2024_

### Official Review · Reviewer_dYDc · 2023-10-30

**Soundness:** 2 fair
**Presentation:** 1 poor
**Contribution:** 2 fair
**Rating:** 3
**Confidence:** 2

**Summary:**

The paper delves into set-based learning, emphasizing the importance of Graph Neural Networks (GNNs) and DeepSets for handling complex datasets. The authors introduce "Fishnets", an aggregation strategy tailored for Bayesian inference and graph aggregation. This strategy showcases adaptability, resilience, and superiority in capturing Bayesian information. Moreover, when integrated with GNNs, Fishnets enhances performance and efficiency. This research positions Fishnets as a potential game-changer in deep learning and network science.

**Strengths:**

A large number of experiments are carried out to prove the effectiveness of the method.

**Weaknesses:**

1. The logic of the introduction part is illogical and too short. The reader does not understand what problem this article is going to solve and what the shortcomings of existing methods are. The authors do not emphasize the main contribution of this article and what are the differences from existing methods.
2. "Fisher" is mentioned frequently in the paper. The author should add a "preliminary" section to introduce what Fisher are to make it easier for readers with different backgrounds.
3. There exist a large number of grammatical errors and spelling errors in the article, such as:

- In "GNNs can acheive state-of-the-art", "acheive" should be "achieve".
- In "On one hand, frequentist analyses", it's better to use "On the one hand" for clarity.
- In "In a deep learning context graph neural networks (GNNs) rely", a comma is needed after "context".
- In "for predictive or regression tasks", consider using "for either predictive or regression tasks" for clarity.
- In "Up until now, graph aggregation schemes", the comma after "now" is unnecessary.
- In "This paper is organised as follows:", "organised" might be considered a British spelling. You might want to use "organized", the American spelling.
- In "In this paper we built up", a comma is needed after "paper".

**Questions:**

please refer to the Weaknesses.

---

> ### Author Response · Authors · 2023-11-22
>
> Thank you for your thorough response and questions !
>
>
> **Updates to the paper:** We have revised the introduction in Section 2 to clarify the rigorous theoretical foundation, based on the likelihood principle, upon which we base our claim of information optimality. We include a detailed treatment of the information inequality and the Cramer-Rao bound (with new proof in the appendix), and then proceed to set-based likelihoods and Fishnets embedding. We additionally provide a new, more in-depth empirical analysis of Fishnets aggregation on three OGB graph datasets (for graph- and node-level prediction). Replacing their aggregators with Fishnets we are able to reproduce or exceed the SOTA benchmarks of Li et al’s DeeperGCN paper but with far fewer GNN layers and learnable parameters. Finally, we improved the organisation and flow of the paper from theory to Bayesian information saturation to GNN prediction.
>
> **Response to comments:** We have revised our introduction section and explicitly defined information optimality and the Fisher information matrix. We also include an Appendix in which we prove information saturation with knowledge of the score function.
>
> Grammatical and spelling errors have been addressed.

---

### Official Review · Reviewer_XpEu · 2023-10-31

**Soundness:** 3 good
**Presentation:** 2 fair
**Contribution:** 2 fair
**Rating:** 3
**Confidence:** 4

**Summary:**

This paper proposes a method of aggregation based on Fisher information. The authors show significant problems in existing aggregation methods such as mean or softmax, and conduct several experiments to show that the proposed method does not suffer like existing methods.

**Strengths:**

- The paper shows significant problems with using existing aggregation methods.

- The experiments seem to show that the proposed method does not suffer from the same issues as using other aggregation methods

**Weaknesses:**

- The experiments are not very convincing. While I do appreciate the simple examples, I think that additional comparisons on real world benchmarks should be conducted to truly show the effectiveness of the proposed aggregation scheme, also given that the authors say the proposed mechanism can be implemented as a 'drop in' aggregation function. Therefore it should be shown on multiple datasets and multiple GNN backbones.

- Also, in the experimental results on the ogb protein dataset, the authors should compare with more methods. The results of other methods are significantly better than just using GCN.

- It is not clear what is the computational cost of the method. It is not discussed and not reported in runtimes.

- The authors say both in title and in abstract that the method is optimal, but it is not proven anywhere, and again the experimental results are lacking.

**Questions:**

- The method itself seems to be more general than just to be applied to GNNs. Did you try to use it for other types of neural networks? what are the limitations of the method?

---

> ### Author Response · Authors · 2023-11-22
>
> Thank you for your thorough response and questions !
>
> **Updates to the paper:** We have revised the introduction in Section 2 to clarify the rigorous theoretical foundation, based on the likelihood principle, upon which we base our claim of information optimality. We include a detailed treatment of the information inequality and the Cramer-Rao bound (with new proof in the appendix), and then proceed to set-based likelihoods and Fishnets embedding. We additionally provide a new, more in-depth empirical analysis of Fishnets aggregation on three OGB graph datasets (for graph- and node-level prediction). Replacing their aggregators with Fishnets we are able to reproduce or exceed the SOTA benchmarks of Li et al’s DeeperGCN paper but with far fewer GNN layers and learnable parameters. Finally, we improved the organisation and flow of the paper from theory to Bayesian information saturation to GNN prediction.
>
> **Response to comments:** We have incorporated more real-world benchmark datasets, which show improvement in performance with fewer learnable parameters (see new Section 5.1).
>
> We wanted to keep the study limited to highlight the advantage of Fishnets aggregation compared to Li et al’s learned softmax aggregation. We do not compare to other architectures since we wish to highlight the improvement of the aggregation scheme with other hyperparameters held fixed.
>
> We report improvement in training epoch reduction to optimality, as well as smaller networks (fewer learned parameters to perform gradient descent on).
>
> Optimality is now discussed in depth in Section 2 and Appendix A.

---

### Official Review · Reviewer_ZQ1B · 2023-10-31

**Soundness:** 3 good
**Presentation:** 1 poor
**Contribution:** 3 good
**Rating:** 5
**Confidence:** 4

**Summary:**

Paper proposes a way to aggregate a set of variable-sized feature vectors into a fixed vector. This is essential for set-based inputs (where the number of items in the set is variable) and graph neural networks (where number of neighbors changes per node). Paper models  data distributions, $p(d | \theta)$, using the Fisher information matrix $F = \sum_{d \in dataset} \nabla_\theta p(d|\theta) \cdot \nabla_\theta p(d|\theta)^T$ where the derivatives (**I think**) are calcuated at a given value of $\theta$. Then, the value of $\theta$ is iteratively updated (**I thnk**) by repeatedly updating with  $F^{-1} t$, where $t = \nabla_\theta \sum_{d \in dataset}   p(d|\theta) $. Paper claims that this yields an information-optimal representation of data.


# Update

I read the author's response and updated manuscript. I will keep my score as-is

## Response

Thank you for your hard work and quick turn-around.

Unfortunately, the paper still seems incomplete in my opinion. I will unlikely change my rating at this point. The following are points you may consider for future versions (e.g., resubmissions)

* Add some prelim section.
* Equation 6: Please be clear on **iterate**. I expect to see something like parameter at $t$ versus $t - 1$,  etc.
* Above Eq.12: it is not clear what is meant by "draw data-parameter pair" -- does this mean, randomly draw subset of data, and on it, train a neural network (to convergence?)
* Please avoid using "$\cdot$" on LHS of Equations 14 & 15.

**Strengths:**

* **Direction**: The paper addresses an important general direction: Combining variable-length (orderless) information is applicable to many tasks, including graphs (social networks, biological networks, user-product interactions, etc), sets, or ordered sequences (text or video) if positional sinusoidal embeddings are added. Their *subdirection*, i.e., learning a neural net that output summary statistics for distributions, is also important and impactful (e.g., Neural Statistician, https://arxiv.org/abs/1606.02185) with many possible combinations: Given a set, whats the probability of an item given others in the set.

* **Practical**  (1)The method computes per-example a gradient term $J_i$ (and its outer-product $J_i J_i^T$), which can be computed in parallel. Then, the sum across all examples. (2) It can be used as drop-in replacement for GNN training. (3) Method trains fast and doesnt use many parameters

* **Pointing out primary weakness of related ( / previous) work.**: Computing $F^-1$ (where F is fisher matrix) gives "asymptotic optimality".

**Weaknesses:**

* The primary weakness is that the paper does not stand on its own. While nicely written and makes important contributions (e.g., practical &  theoretical), the missing (preliminary) information makes this paper difficult to understand and reproduce, unless the reader goes in-depth into related work. I will detail this and ask the authors some clarification questions.

* "**optimality**" is mentioned a few times. However, it is not clear from the text: "**optimal, from what sense?**". In my opinion, there needs to be a *clearly-stated (i) objective function or (ii) inequality, and showing that the method either recovers the (i) (unique) optimal or (ii) satisfies the inequality with equality. Currently, this discussion is skipped.

* Experiments. The only real dataset is ogbn-protein. If you already integrated with PyG and have ogbn training setup, wouldn't it be straight forward to rerun your code while changing only the dataset name?

**Questions:**

# Please be more exact

* Define $\theta_{fid}$.

* What is the iteration in "(iteratively)" mentioned before Equation 3?

I think that all of the above should be incorporated in the paper

In Equation (9), please list-down the parameters which the objective function is optimized w.r.t., e.g.

$$\mathcal{L}(w_t, w_F)$$

The reason I am asking for this, because it is not immediately obvious if $\theta$ is a constant, though it is clear that $\hat{\theta}_{NN}$ is parameterized by $(w_t, w_F)$. Further, it is not clear if $\theta_{fid}$ is always fixed -- if not, please list in LHS of Eq.9.

# Questions


1. What is $\theta_{fid}$? what is the what is the **iterate** in ``(iteratively)''  mentioned before Equation 3? Do the authors imply that $\theta_{fid}$ is the previous iteration value and $\hat{\theta}^{MLE}$ is the next iteration?

2. What is $\theta$ in Eq.9?

3. In equation (9), $\theta_{NN}$ is defined in terms of $F_{NN}$ and $t_{NN}$ (see Eq.8).

4. What is the space / dimensionality of $d$ (the input examples). You can be general. Specifically, do Equations 11 and 12 have a range equal to space of $d$, or do they output scalars? -- this question is tied to next.

5. Eq.11 & Eq.12: Can you clarify the input argument? Is $\beta$ a vector or scalar?

If the rebuttal answers gives me a homework to "read the referenced papers well", then perhaps it is very much worth making a "Preliminaries" section to define missing expressions.

---

> ### Author Response · Authors · 2023-11-22
>
> Thank you for your thorough response and questions !
>
> **Updates to the paper**: We have revised the introduction in Section 2 to clarify the rigorous theoretical foundation, based on the likelihood principle, upon which we base our claim of information optimality. We include a detailed treatment of the information inequality and the Cramer-Rao bound (with new proof in the appendix), and then proceed to set-based likelihoods and Fishnets embedding. We additionally provide a new, more in-depth empirical analysis of Fishnets aggregation on three OGB graph datasets (for graph- and node-level prediction).  Replacing their aggregators with Fishnets we are able to reproduce or exceed the SOTA benchmarks of Li et al’s DeeperGCN paper but with far fewer GNN layers and learnable parameters. Finally, we improved the organisation and flow of the paper from theory to Bayesian information saturation to GNN prediction.
>
>
> **Response to Comments:** In our new introduction section (Section 2) we clarify our notion of information optimality and show how our neural embeddings (the score $t(d_i)$) and weights ($F(d_i)$) are learned as a function of the data and underlying implicit likelihood. Please refer to this section for clarification on the method (no gradients at a value of $\theta$ are needed).
>
> Weaknesses:
> W1: the new introduction section and accompanying appendix should clarify the practical and theoretical components that were missing from the original submission.
>
> W2: Optimality is now described in Section 2 (proof in Appendix A)
>
> W3: We have now produced more OGB experiments, with similar levels of improvements once Fishnets is adopted.
>
> Q1: $\theta_{\rm fid}$ has been renamed $\theta_*$ and defined in the update
>
> Q2: Please see Section 2 and Appendix A
>
> Q3: this has been clarified in Section 2
>
> Q4: $\textbf{d}_i \in \mathbb{R}^N$; $\textbf{t}_i \in \mathbb{R}^{n_p}$
>
> Q5: Eqs 11&12: the input to the softmax aggregation is either a data vector $\textbf{d}_i \in \mathbb{R}^N$,
>
> or some neural network embedding $ q_i \in \mathbb{R}^{n_{\rm h}} $. This has been clarified in both cases.

---

### Official Review · Reviewer_KY3Y · 2023-11-01

**Soundness:** 2 fair
**Presentation:** 1 poor
**Contribution:** 2 fair
**Rating:** 3
**Confidence:** 4

**Summary:**

This paper proposes Fishnets, an aggregation strategy for learning information-optimal embeddings for sets of
data for both Bayesian inference and graph aggregation.
The methods contribute to the general set aggregation problems.

**Strengths:**

- Good methodological motivations regarding optimal aggregation of independent heterogeneous data

**Weaknesses:**

- The paper is poorly structured. The Introduction is too short to introduce and motivate the problem. The method and related work section is too limited. The experimental section appears on page 3. Overall, the paper does not appear to be properly written.
- The Bayesian Information saturation experiments are detached from graph neural network aggregation experiments. Why they should be separately discussed? And why the GNN aggregation experiments cannot share the same analysis from the Bayesian Information saturation experiments?
- Only experimenting on OGBN-Proteins seems to be too limited. Experiments on more graph datasets are needed.
- Only using deep GNNs is a weird setting. Why not experiment with shallow GNNs? The paper should at least include results regarding shallow GNNs for comparison.

**Questions:**

Why the GNN aggregation experiments cannot share the analysis from the Bayesian Information saturation experiments?

---

> ### Author Response · Authors · 2023-11-22
>
> Thank you for your response and questions !
>
> **Updates to Paper:** We have revised the introduction in Section 2 to clarify the rigorous theoretical foundation, based on the likelihood principle, upon which we base our claim of information optimality. We include a detailed treatment of the information inequality and the Cramer-Rao bound (with new proof in the appendix), and then proceed to set-based likelihoods and Fishnets embedding. We additionally provide a new, more in-depth empirical analysis of Fishnets aggregation on three OGB graph datasets (for graph- and node-level prediction).  Replacing their aggregators with Fishnets we are able to reproduce or exceed the SOTA benchmarks of Li et al’s DeeperGCN paper but with far fewer GNN layers and learnable parameters.
> Finally, we improved the organisation and flow of the paper from theory to Bayesian information saturation to GNN prediction.
>
> **Response to Comments:** The Bayesian Information saturation section is detached from the graph neural network aggregation experiments because in the latter we wished to compare to a ground truth inference, e.g. the HMC sampler, which is not possible for graph aggregation tasks because we do not have access to sampling distributions for the ogbn-proteins dataset features. The noisy edge protein analysis demonstrates how GNN point estimators can be made robust to noise.
>
> We have incorporated more OGB dataset comparisons (Section 5.1). We show that we obtain same- or better results with shallow Fishnets GCNs in these benchmark cases.

---

### Official Review · Reviewer_BUmE · 2023-11-01

**Soundness:** 4 excellent
**Presentation:** 3 good
**Contribution:** 4 excellent
**Rating:** 6
**Confidence:** 2

**Summary:**

In this paper, the authors propose Fishnets, a strategy that can better learn embeddings for sets of data, used in DeepSets environments or in graphs as well. The method is statistically-backed with solid motivation and organization behind the writing, and achieves solid performance boosts when it comes to the empirical tests.

**Strengths:**

The strengths of this paper lie in its rigor, novelty, and strong performance. The Fishnets strategy is well-motivated in the second section, with both intuition and theory combined. In addition, because of this it seems to be relatively novel compared to similar works, which also helps Fishnets achieve great performance empirically, no matter the dataset/task.

**Weaknesses:**

The main weaknesses of the paper lie in not that many empirical comparisons, as well as a slight decline in organization as the paper goes on. For example, there could be better organization between sections 4 and 5; perhaps differentiating a little bit more or talking about the experiments in the introduction. The sections do not seem to have a reasonable ordering to them, but rather are jumbled with one another, which also makes experiments much harder to conduct, given that there are so many applications and so many claims being made. For example, one key paradigm, being the graph learning, is only given one dataset to work with. For such an important aspect of the work, it would be expected to include performance on multiple graph-like datasets.

**Questions:**

1. The authors mention faster training time in multiple occurrences, but to the best of my knowledge data on computation time is not included in the paper. Would it be possible to shed a little bit of light on how much the improvement is on that front?
2. Why was the proteins dataset chosen in the first place over any other potential graph benchmark?

---

> ### Author Response · Authors · 2023-11-22
>
> Thank you for your response and questions !
>
> **Updates to paper:** We have revised the introduction in Section 2 to clarify the rigorous theoretical foundation, based on the likelihood principle, upon which we base our claim of information optimality. We include a detailed treatment of the information inequality and the Cramer-Rao bound (with new proof in the appendix), and then proceed to set-based likelihoods and Fishnets embedding. We additionally provide a new, more in-depth empirical analysis of Fishnets aggregation on three OGB graph datasets (for graph- and node-level prediction).  Replacing their aggregators with Fishnets we are able to reproduce or exceed the SOTA benchmarks of Li et al’s DeeperGCN paper but with far fewer GNN layers and learnable parameters. Finally, we improved the organisation and flow of the paper from theory to Bayesian information saturation to GNN prediction.
>
> **Response to Comments:**
>
> Q1: Faster training time here refers to fewer learning (gradient descent) epochs needed to achieve convergence of the GNN model. We indicate this improvement by showing test ROC-AUC curves in the ogbn-proteins focus section. We additionally indicate improved benchmark comparisons (see Section 5.1) with drastically fewer learnable parameters than Li et al’s best GCN models.
>
> Q2: We initially chose to work with the ogbn-proteins dataset because the prediction task consisted of aggregating weighted edges (association scores) between proteins (nodes) in the graph. These association scores could be modelled with noise akin to the linear regression and BHM models in previous sections to demonstrate the robustness of the Fishnets aggregation to changing noise because the Fisher weighting scheme explicitly re-weights embeddings in a heterogeneous manner as a function of data (and in the linear regression and noisy proteins case, the $\sigma$ and $N$ noise amplitudes).

---

> > ### Comment · Reviewer_BUmE · 2023-11-23
> >
> > Thank you for your answers and edits.
> >
> > I don't believe there to be any significant/major changes to the paper to warrant an increase/decrease in score, so I will retain my score for now.

---

### Official Review · Reviewer_wjmC · 2023-11-01

**Soundness:** 2 fair
**Presentation:** 3 good
**Contribution:** 3 good
**Rating:** 5
**Confidence:** 4

**Summary:**

The paper proposes an aggregation strategy for sets of data that aims to learn “information-optimal” embeddings. The approach resides around training neural networks to learn the score and Fisher information matrix for individual data points. This permits the construction of optimal embeddings by aggregating the scores and Fisher matrices and applying the Fisher scoring method iteratively. The paper reports the results of multiple experiments that demonstrate the potential applications of the methodology and and highlight its strengths - robustness to distribution shift, ability to saturate Bayesian information content, capability to take into account uncertainty.

**Strengths:**

S1: The paper proposes a principled approach for aggregation of set data to construct improved embeddings.

S2. Several experiments demonstrate the potential of the proposed methodology including (i) its ability to saturate Bayesian information content; (ii) its robustness to distribution shift; and (iii) its usage as an aggregation technique in GNNs.

**Weaknesses:**

W1: The presentation of the method could be improved. In particular, the key concepts of “information optimal” and “information saturation” are not clearly defined. Section 2 first explains that the Fisher scoring method can be used to “(iteratively) form a maximum likelihood estimator” and then transitions to a “pseudo-MLE”. Alsing & Wandelt (2018) provide a considerably more complete and clearer discussion. This paper should aim to be more precise. For example, it’s not clear what “(iteratively)” means here – Alsing & Wandelt spell it out that it converges in the limit and explain that after k iterations it provides a (clearly defined) quasi maximum-likelihood estimator. The paper doesn’t explain the transition to a “pseudo-MLE” later in the section. It doesn’t explain the concept of “saturating” the Fisher information; it doesn’t provide the Cramer-Rao bound. One could argue that some of this might be considered necessary background knowledge, but I would suggest that the paper should be accessible to practitioners who are interested in learning about a new aggregation method and perhaps do not have an information theoretic background.

W2: Although the experiments provide illustrations of how the approach can prove beneficial, none of them constitutes a thorough experimental study that demonstrates that the methodology is superior for a given application. In each case, the proposed approach is compared to one baseline, for one dataset or model. This is sufficient to illustrate the potential of the method, but it does not provide compelling evidence that the method can be a state-of-the-art choice for a particular problem.

An example is the GNN aggregation experiment. Rather than providing a systematic and complete study over multiple datasets and comparing against multiple baselines to demonstrate how the proposed method acts as the best aggregator (in some sense) the provided experiments are for a single dataset, with essentially two variants of single baseline as comparison points. The selected baseline does not achieve close to the state-of-the-art performance – as asked below, it’s not even clear that the selected baseline is configured to achieve its best performance. So these experiments allow us to conclude that for one dataset the proposed method outperforms one questionably-configured baseline that is itself well behind the state-of-the-art (although one could make an argument that the leaders above it are not “pure” GNNs and are achieving outperformance through processing steps that are not related to graph learning or aggregation). The experiments definitely do not establish that the proposed aggregator can be reliably combined with multiple different GNN architectures and employed confidently for a new dataset.

It's not essential that the paper provide such evidence for every application of the proposed method. Indeed, it is good to see the versatility via illustrations of how it could be employed in a range of settings. However, the paper would be much stronger if it provided compelling evidence for at least one of the suggested applications. The paper does not make a strong theoretical contribution, although the proposed technique is well-founded theoretically. As such, the bar for experimentation is higher; one expects convincing empirical evidence to support the claims.
W3. Related work: The Deep Sets paper was published in 2018 and has been cited over 2000 times. While many of these papers involve applications of the method, many others propose extensions and examine the aggregation process. As just one example, [R1] examines learnable recurrent aggregation functions. A related work section that discusses two papers and then cites three other examples with a single sentence (all dating from 2020 or earlier) is inadequate. This section does not cite any papers published in 2021-2023, making it highly questionable whether the paper is correctly positioning the proposed method in the context of existing research on the topic.

[R1] Soelch, Maximilian, et al. "On deep set learning and the choice of aggregations." Artificial Neural Networks and Machine Learning–ICANN 2019.

**Questions:**

Q1. Why does the AUC-ROC performance for the baseline differ so much from the value reported in Li et al. 2020 (where the best performing model is 0.855 for a 28 layer network)? Do the experiments avoid residual connections for some reason? (The description “out-of-the-box 28-layer GCN” is unclear to me). Li et al. explicitly state “residual connections significantly improve the performance of deep GCN models. PlainGCN without skip connections does not gain any improvement from increasing depth.”

Q2. “Explicitly learning the $F^{−1}$ weights in addition to the score allows us to achieve 1) asymptotic optimality” – can the paper concretely define the nature of the optimality? is there a simple proof? Or is it considered obvious? Presumably there a reliance on a universal function approximation result for the neural networks learning the score and the Fisher? Does this combine obviously with the asymptotics of the Fisher method?

---

> ### Author Response · Authors · 2023-11-22
>
> Thank you for your detailed review !
>
>
> **Updates to Paper:**
> We have revised the introduction in Section 2 to clarify the rigorous theoretical foundation, based on the likelihood principle, upon which we base our claim of information optimality. We include a detailed treatment of the information inequality and the Cramer-Rao bound (with new proof in the appendix), and then proceed to set-based likelihoods and Fishnets embedding.
> We additionally provide a new, more in-depth empirical analysis of Fishnets aggregation on three OGB graph datasets (for graph- and node-level prediction).  Replacing their aggregators with Fishnets we are able to reproduce or exceed the SOTA benchmarks of Li et al’s DeeperGCN paper but with far fewer GNN layers and learnable parameters. Finally, we improved the organization and flow of the paper from theory to Bayesian information saturation to GNN prediction.
>
> **Response to Comments:**
>
> Answers to Questions:
>
> Q1: We have updated the codebase to compare Li et al’s best-performing models to our aggregation technique. We first do a drop-in replacement study on three OGB datasets with Li et al’s DeeperGCN code and achieve similar results with their baseline models before proceeding with our Fishnets modifications.  We revised our initial ogbn-proteins study into a focus study. We used a stripped-down version of Li et al’s codebase initially in order to modify the raw data during training in the noisy edge case. We provide our benchmark models within this training and preprocessing scheme to provide a benchmark against which to compare the noisy setting.
>
> Q2: This optimality guarantee (under the likelihood principle) has been explicitly provided in Section 2 with proof in Appendix A.

---

> > ### Comment · Reviewer_wjmC · 2023-11-23
> > **Acknowledgement of the response**
> >
> > Thank you for the response to my review and the changes that have been made to the paper. While the paper has improved substantially, I think there is still the need for improved clarity. Overall, this seems like a very clever and promising idea, but the paper needs a careful and thorough revision before it is ready for publication.
> >
> > The experimentation has been extended to include more datasets, which is good, but there is still the limitation to a single baseline for the GNN aggregation, and the experimental details are hard to follow.
> >
> > My concern regarding the positioning with regard to more recent related work was not addressed - the paper still includes very limited discussion of other aggregation strategies, and does not discuss any related work published since 2020.
> >
> > The meaning and claim of asymptotic optimality would be clearer with a formal theorem/lemma/corollary and an associated proof. Currently, it seems that the discussion in Appendix A is connected to Section 2.1. Both of these sections develop the material using the true Fisher and the true score. But the claim of optimality is being made for the technique in Section 2.3. In that section, the Fisher and the score are learned (estimates). There doesn't seem to be a sufficiently concrete linkage between the optimality (based on true Fisher and score) in Section 2.1 & Appendix A, and the claimed optimality for learned/estimated Fisher and score in Section 2.3. Phrases like "Provided the embeddings ... are learned sufficiently well" are a concern, because there is no specification of what "sufficiently well" means. This is where a formal statement of the result would be helpful, because it would be made clear exactly what is being assumed.

---

> > > ### Author Response · Authors · 2023-11-23
> > > **Reply to Acknowledgement**
> > >
> > > Thank you for your follow-up response ! What should be emphasised is that the *form* of the aggregation is optimal under the likelihood principle.
> > >
> > > Here we model members of a set (be it a set of data or graph node neighbourhood) as members of an *implicit* probability distribution (likelihood) $p( \\{ d_i \\} | \theta)$, where $\theta$ is some quantity of interest, be it a parameter of a statistical model or some latent representation within a graph network. The optimality with respect to a *true* score or Fisher can be demonstrated empirically against a ground-truth in analytically-known cases, like the linear regression validation case. However, for implicit (intractable) likelihoods, we rely on the form of the aggregation (score embedding weighted by a the inverse-Fisher weight matrix) to guide our neural network architecture design to improve aggregated information capture.
> > >
> > > We agree that the wording "sufficiently well" might be misleading and imply that there is some guarantee that the true score and Fisher are being learned in every case. We will consider changing this to "descriptive neural embedding" for clarity.

---

### Official Review · Reviewer_D4AK · 2023-11-02

**Soundness:** 2 fair
**Presentation:** 1 poor
**Contribution:** 3 good
**Rating:** 5
**Confidence:** 3

**Summary:**

This paper proposes a scalable aggregator Fishnets for sets and graphs from the perspective of MLE. Fishnets can approximate score and Fisher matrix from data with arbitrary size with neural networks. It has comparable performance with DeepSets and Softmax Aggregation while using fewer parameters.

**Strengths:**

1. This model has good scalability as it simply keeps the summation form regardless of the size of data points.
2. The model architecture is simple and it has fewer parameters than the other counterparts.

**Weaknesses:**

1. The model requires to calculate the inverse of Fisher matrix in each iteration of the estimator. I am concerned with the computation complexity. I believe the complexity analysis and runtime comparison with other counterparts are needed.
2. The proposed model has limited flexibility, as it is only permutation invariant. While its counterpart DeepSets have both invariant and equivariant formulations.
3. The proposed models are only compared with the mentioned two counterparts, which are not enough to demonstrate the effectiveness. More extensive experiments are needed.
4. The clarity of this paper needs to be improved. Many symbols have unexplained meanings and shapes.

**Questions:**

1. What makes the Fishnets robust to noise?
2. In section 5, why does the experiments only perform on deep GNNs? Edge aggregation can be also performed on shallow GNNs, such as GAT [1].

[1] Veličković, Petar, et al. "Graph attention networks." arXiv preprint arXiv:1710.10903 (2017).

---

> ### Author Response · Authors · 2023-11-22
>
> Thank you for your response and questions !
>
> **Updates to Paper:** We have revised the introduction in Section 2 to clarify the rigorous theoretical foundation, based on the likelihood principle, upon which we base our claim of information optimality. We include a detailed treatment of the information inequality and the Cramer-Rao bound (with new proof in the appendix), and then proceed to set-based likelihoods and Fishnets embedding. We additionally provide a new, more in-depth empirical analysis of Fishnets aggregation on three OGB graph datasets (for graph- and node-level prediction).  Replacing their aggregators with Fishnets we are able to reproduce or exceed the SOTA benchmarks of Li et al’s DeeperGCN paper but with far fewer GNN layers and learnable parameters. Finally, we improved the organisation and flow of the paper from theory to Bayesian information saturation to GNN prediction.
>
> **Response to Comments:**
> To respond to your questions, we refer you to our revised Section 2. Fishnets is robust to heterogeneous noise because we explicitly parameterise the inverse-Fisher weighting scheme as a learned function of the data (which in the linear regression and noisy proteins cases includes estimates of the noise variance).
>
> We additionally revised our GNN section to include far more examples of the Fishnets aggregation. We elected to use the GCN framework due to many readily-available examples, as well as to show that the Fishnets aggregation could improve performance and reduce learnable parameters as a drop-in replacement.

---

### Meta-Review · Area_Chair_6Md4 · 2023-12-05

**Metareview:**

The reviewers are in agreement that the manuscript is below the bar for acceptance at this time, especially in terms of its lack of comparison of alternative aggregation strategies in theory or practice.

**Justification For Why Not Higher Score:**

NA

**Justification For Why Not Lower Score:**

NA

---

### Decision · Program_Chairs · 2024-01-16

Reject